# Importance of Interfacial Adhesion Condition on Characterization of Plant-Fiber-Reinforced Polymer Composites: A Review

**DOI:** 10.3390/polym13030438

**Published:** 2021-01-29

**Authors:** Ching Hao Lee, Abdan Khalina, Seng Hua Lee

**Affiliations:** Institute of Tropical Forestry and Tropical Products (INTROP), Universiti Putra Malaysua (UPM), Jalan UPM, Serdang 43400, Malaysia; lee_seng@upm.edu.my

**Keywords:** interfacial adhesion, plant fiber, methodology, characterization, polymer composites

## Abstract

Plant fibers have become a highly sought-after material in the recent days as a result of raising environmental awareness and the realization of harmful effects imposed by synthetic fibers. Natural plant fibers have been widely used as fillers in fabricating plant-fibers-reinforced polymer composites. However, owing to the completely opposite nature of the plant fibers and polymer matrix, treatment is often required to enhance the compatibility between these two materials. Interfacial adhesion mechanisms are among the most influential yet seldom discussed factors that affect the physical, mechanical, and thermal properties of the plant-fibers-reinforced polymer composites. Therefore, this review paper expounds the importance of interfacial adhesion condition on the properties of plant-fiber-reinforced polymer composites. The advantages and disadvantages of natural plant fibers are discussed. Four important interface mechanism, namely interdiffusion, electrostatic adhesion, chemical adhesion, and mechanical interlocking are highlighted. In addition, quantifying and analysis techniques of interfacial adhesion condition is demonstrated. Lastly, the importance of interfacial adhesion condition on the performances of the plant fiber polymer composites performances is discussed. It can be seen that the physical and thermal properties as well as flexural strength of the composites are highly dependent on the interfacial adhesion condition.

## 1. Introduction

Every year, hundreds of studies regarding plant-fiber-reinforced polymer composites were published in various journals and the trend has been increased exponentially [1]. The application of plant fibers in polymer composites have drawn attention of many industry manufacturers [2]. Urgent call for ameliorating environmental impacts by reducing energy consumption and embedding biodegradability but retaining reasonable performances are the major driving forces for the development of plant-fiber-reinforced polymer composites. In comparison with synthetic fibers, plant fibers offered multiple advantages such as light weight, biodegradability, low price, and life-cycle superiority. However, some drawbacks of plant fibers imposed challenges to the development and application of plant-fibers-reinforced polymer composites. With the collaboration between researchers, properties of the plant-fiber-reinforced polymer composites is enhanced to a much greater extent.

Matrixes are generally a homogeneous and monolithic material in which fiber and/or fillers system of a composite is embedded. It is completely continuous and provides a medium for binding and holding reinforcements together into a solid structure. The main purpose of matrixes is to offer protection and transfer loads to the reinforcement fillers [3]. On the other hand, plant fibers are held in place by the matrix resin, sustaining under high strength and enhancing performances of composites with almost zero cost. However, cooperation between plant fibers and matrix in a composite system relies on its interface conditions. 

Interface conditions are the prime factor of determining the properties of plant fiber polymer composites. This region experiences different thermal expansion during thermal processing and it acts as a barrier between the two distinct materials that differed in terms of physical and chemical properties. A composite with bad interface may find significant deterioration in mechanical and thermal properties. Even worse, inferior physical properties was also reported. Fortunately, numerous treatments or compatibilizers have been applied to the composites with the aim of enhancing physical, mechanical, and thermal characterizations. Alkaline treatment is the most widely used treatment on plant fiber composites because of its high cost-effectiveness [4,5,6]. It removes non-cellulosic components on the fiber surface and offering a clean but rough surface for better interfacial adhesion. Coupling agents and compatibilizers are deployed to promote more functional groups to enhance interfacial adhesion between fiber/matrix [7,8,9,10,11]. Through the abovementioned material modifications, researchers could improve conditions of interfacial bonding between the fiber/matrix’s interface, as illustrated in Figure 1.

A good interface condition is facilitated by one or a mixture of four (4) interface linkage mechanisms, namely interdiffusion, electrostatic adhesion, chemical adhesion and mechanical interlocking. Interdiffusion bonding mechanism is governed by wettability. Optimum surface energy and polarity on both fibers and matrix, create permanent adhesion via Van der Waals, covalent, and electrostatic forces. Electrostatic adhesion uses attraction of cations and anions to form the interface. However, limited studies on electrostatic adhesion of plant fiber polymer composite were reported. Besides, improvement of hydrophobicity of the fibers via treatments may increase the chemical adhesion compatibility with hydrophobic matrix. Removal of fiber’s hydroxyl groups and substitutes with hydrophobic chemical bonding could assist to enhance the hydrophobicity of plant fibers. On the other hand, penetration of molten polymer into micron-diameter holes and adhered on irregular, rough fiber surface has created mechanical interlocking support system. This adhesive mechanism does not rely on any chemicals bonding or electrostatic forces, but the polymer acts like multiple hooks anchored on the fiber surface. Rougher fiber surface provides more spots to anchor. 

To quantify the conditions of fiber/matrix interface, multiple analysis methods could be used to understand the interface situations. Thermodynamic analysis applies flowing liquid on fiber surface to analyze the solid–liquid contact angle and in turn, knowing the wettability at the interface. However, heterogenous properties of plant fibers show divers wetting conditions. Inverse gas chromatography (IGC) is a better tool to study dispersive surface energy and thermodynamic of adsorption. Besides, microscopic viewing analysis provides concrete graphical evidence in microns level. Topography of fibers and matrix enable us to predict interface condition and hence composite’s performances. Scanning electron microscope (SEM), transmission electron microscopy (TEM), and atomic forced microscopy (AFM) analysis are conducted frequently to show topography in graphical ways. 

There are three main constituents to make up a single plant fiber. Interface conditions between fibers and matrixes are strongly influenced by their chemical compositions. Therefore, spectroscopic techniques to analyze chemical composition of plant fibers are important. Removal of non-cellulosic components on fiber surface or promotion of extra functional groups have aided to promote better interfacial bonding with the matrix. These can be seen from disappearance or appearance of absorbance peak from Fourier transform infrared spectroscopy (FT-IR). Besides, higher crystallinity index helps to enhance mechanical properties of composite as cellulose component is the best strength tolerator. On the other hand, micromechanical measurements such as single fiber pull-out test analyze the interface conditions without requiring the production of large-scale composite. Depending on the interfacial adhesive conditions, the plant fibers may pull-out, debond, or break. However, fiber fibrillation make fiber turn into finer diameter. This recorded lower load capability for each fiber, increases total fiber surface area for better adhesion in composites. 

Interface adhesive conditions between plant fibers and matrixes are the main factor influencing the physical, mechanical, and thermal properties. Higher density but lower diameter of plant fiber was found upon surface cleaning treatments. This provides better wettability with polymer matrix and formation of voids could be reduced. Lower water absorption shall be observed at the same time. Plant fibers with smaller diameter also increases corresponding aspect ratio (L/d) values. This increment is the most influential parameter influencing the mechanical properties of the composites. However, flexural strength is influenced by interface conditions more than tensile strength. This is because flexural bending loads imply a combined compressive/tensile and interfacial shear stress to the specimens, in which interfacial adhesion plays a fundamental role. 

On the other hand, good interface bonding requires higher amount of thermal energy to break the bonds. The same strong bonding recorded lower energy dissipation by internal friction. Therefore, higher thermal stability and storage modulus retained for the composites. Insertion of plant fibers also offered nucleating site for polymer cold crystallization, shifting higher crystalline temperature. Bonding in the interface prevent sliding of polymer chains, delaying the transition from glassy state to rubbery state, known as glass transition temperature, *T*_g._

The importance of interface adhesion condition for plant-fiber-reinforced polymer composites has not been questioned by all researchers. Understanding the interfacial adhesion mechanisms and its quantitative measurements are useful in characterization study of plant-fiber-reinforced polymer composites. The aims of this review are to provide audiences a comprehensive information on plant fiber/matrix interface, from the interface fundamental, frequently used analyzed techniques and influences on composite properties. 

## 2. Natural Plant Fibers

Natural fiber is not a strange term in the current decade. It has been advertised as an alternative material to non-degradable materials by the companies supporting green materials, to fight global warming and supporting local social economic [13,14,15]. This has contributed to the discovery of more renewable natural resources and join the competitive natural fiber markets. 

Among the big families of natural fibers, plant fibers are the most extensively developed. The plant fibers are categorized by location where the fiber was obtained like stem, leaf, seed, and grass [16]. Plant fibers are generally inexpensive byproducts, bestowed with high strength-to-weight ratio, volume-to-weight ratio, and excellent biodegradability. Detailed advantages and disadvantages of plant fibers are listed in Table 1. These gifted properties made them comparable to synthetic fibers [17]. However, drawbacks of plant fibers were found and reported by previous reviews and studies [4,18,19,20]. The most unfavored characteristic of plant fibers is their hydrophilic nature, which made them incompatible with hydrophobic polymer. Poor interface adhesion is usually observed in the micrographs of plant-fibers-reinforced polymer composites and followed by weakened properties. 

The properties and adhesion ability of the plant fibers are attributed by its chemical compositions. There are three main chemical constituents in plant fibers, namely cellulose, hemicellulose, and lignin components. Cellulose is responsible for strength tolerator, its hydroxyl groups form strong molecular bonding with polymer at interface layer, regulating superior load transfer mechanism. Hemicellulose component dominant on thermal degradation, moisture absorption, and biodegradation of the fiber as it shows the least resistance. On the other hand, lignin is thermally stable and is greatly accountable for the UV degradation. Table 2 shows the chemical compositions of the frequently used plant fibers. Fiber treatments are commonly applied to modify fiber chemical compositions in order to achieve better interface adhesion [21,22].

**Table 1 polymers-13-00438-t001:** Detailed advantages and disadvantages of plant fibers [23].

Advantages	Disadvantages
Less expensive	Lower mechanical properties (especially impact strength)
Lower weight	Higher moisture absorption
Higher flexibility	Lower durability
Renewable	Poor fire resistance
Biodegradable	Variation in quality
Good thermal and sound insulation	Restricted maximum processing temperature
Eco-friendly	Poor microbial resistance
Nontoxic	Low thermal resistance
Lower energy consumption	Demand and supply cycles
No residues when incinerated	
No skin irritations	

Looking into the structural arrangement of plant fibers, in longitudinal and cross-sectional direction, each fiber was constructed by multilayers of cell walls that are mainly grouped into three sub-layers (S_1_, S_2_, S_3_), and S_2_ layer governs the longitudinal mechanical properties (Figure 2). Then these single fibers are bound with middle lamella as the fiber bundle. Hence, the ideal outcomes from extraction methods or retting processes, was to separate the single fibers from fiber. However, incomplete fiber separation has made the fibers exist as technical and elementary fibers. Various fiber conditions are attributed to different wetting behavior and hence unsatisfied interface adhesive properties may be observed. 

Besides, plant fiber diameter is important to predict a composite’s strength performance. Larger diameter fiber reinforcement is observing to have lower tensile strength, yet lower diameter can be refined by applying fiber treatments or compatibilizers [25]. Small diameter provides higher surface area-to-weight ratio to secure well-dispersion and proper wetting on fibers. This creates a strong interface to regulate superior load-transfer mechanism. Several types of interface mechanisms are discussed in the next section.

**Figure 2 polymers-13-00438-f002:**
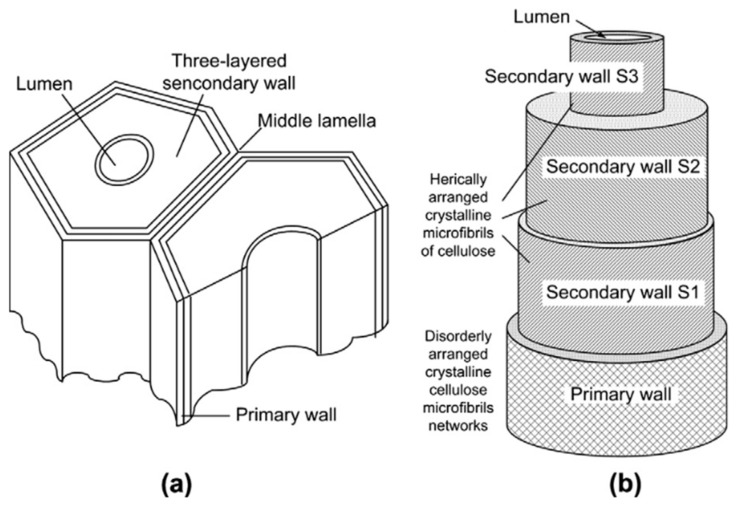
(**a**) Cross section cell arrangements and (**b**) structural of single fiber. Reproduced with permission from [26]. Copyright 2012 Elsevier.

## 3. Interface Mechanisms

For the plant fiber polymer composites, properties are dominant by reinforcing plant fibers, matrix, and most importantly interface conditions. The interface was considered as an intermediate layer, formed by bonding matrix and fibers, in the thickness of one atom to micron thick. A good interlayer forms strong linkages and enable maximum stress transmission between fibers and matrix, without disruption and, hence showing superior properties. Therefore, the conditions of interface are worth investigating in depth. The fiber/matrix interfacial bonding mechanisms are shown in Figure 3, which include interdiffusion, electrostatic adhesion, chemical adhesion, and mechanical interlocking. Typically, interfacial adhesion is an outcome of these multiple bonding mechanisms. Nevertheless, one of them usually plays a dominant role.

### 3.1. Physical Adhesion

Physical adhesion interface is referred to as the interdiffusion bonding mechanism. Good wettability has governed the condition of this interface, which relied on surface energies and polarities, of both plant fibers and matrix. The surface energy and polarity can be analyzed by the contact angle measurements of solid–liquid interactions, which is further discussed in Section 4.1 [28]. Non-polar waxes found on the fiber surface have relatively lower surface tension than polar components like lignin and fats. Fiber surface treatments or maleated coupling agents can be used to regulate the surface energies and polarities to create better wettability. Once good wetting occurs, permanent adhesion is developed through molecular attractions such as Van der Waals, covalent, and electrostatic. Tran et al. [29] found in their study that alkaline-treated coir fibers have lower surface energies but higher polarity. It is more compatible and has better wettability with polymer that has similar surface energy and resulted in higher work of adhesion. The higher the work of adhesion, the better the composite’s mechanical properties [30,31].

### 3.2. Electrostatic Adhesion

The opposite charges because of which contacting surfaces on plant fibers and polymer matrix are attracted and adhered together are known as electrostatic adhesion. Both anionic and cationic bodies formed an interface, which accounts for the adhesion of the two constituents of composite. In the chemical and physical interactions, adhesion property of surfaces is a consequence of interatomic and intermolecular surface forces including electrostatic forces. Atomic force microscope was used to create 3D images of composite topography and investigate its electrostatic adhesion conditions [32]. However, authors in this review paper failed to identify any electrostatic adhesion study on plant-fiber-reinforced polymer composites, and only mentioned briefly the physical adhesion. This may be due to the fact that plant fibers are difficult or not preferred to be processed into a more ionic state. Some studies have reported the integrating electrostatic adhesion to composite structures reinforced with synthetic fibers (carbon or glass fibers) to strengthen the interface [33,34,35]. Electrostatic discharge treatment on polymer fibers or electrostatic fibers by electrospinning process could incorporate electrostatic adhesion to its interface and consequently could provide significant value-added functionality to the composites [36,37]. 

### 3.3. Chemical Adhesion

Chemical bonded interface is the most widely discussed in plant fiber polymer composites. Chemical modifications could be done on both fiber and matrix in order to gain higher intensity of chemical bonding sites. Improved hydrophobicity of fibers could increase the adhesive compatibility with hydrophobic matrix and this could be done by removing fiber’s hydroxyl groups and substitutes with hydrophobic chemical bonding. Figure 4 shows the chemical modification treatments on the fiber’s surface. The details of the chemical reactions have been reviewed [38]. The destruction of hydroxyl groups on fibers prohibited the attraction of water moisture. Hence, improved fiber hydrophobicity and minimized phenomena of swelling-to-crack could be observed. Cracked composite receives lower loads since load transferring mechanism is forced to end and concentrates on the cracking spots [39].

### 3.4. Mechanical Interlocking

Penetration of molten polymer into micron-diameter holes and adhering to irregular, rough fiber surface has created mechanical interlocking support system. This adhesive mechanism does not rely on any chemical bonding or electrostatic forces, but the polymer acts like multiple hooks anchored on fiber surface. Rougher fiber surface provides more spots to anchor. Alkaline treatment is one of most frequently used methods to remove non-cellulosic components from the fiber surfaces, offering a clean and rough fiber topography, other than reducing the hydrophilicity for better chemical adhesion mechanism [41,42]. On the other hand, flow of the polymer resin filled into lumens, open pores, and free volumes within the cell wall has restricted shrinking and swelling of fibers, and thereby better dimensional integrity of composites. The wettability as discussed in the above section (physical adhesion) is the crucial factor for flowability of resin on/into the plant fibers. Hence, the mechanical interlocking often provides extra load-bearing capabilities to the interface. 

## 4. Quantifying and Analysis of Interfacial Adhesion Condition

Quantifying and analysis of interfacial adhesion properties is very important to compare or predict the properties of plant-fiber-reinforced polymer composites. Treatment may improve the interface conditions. However, excessive treatment is not beneficial to interface adhesion, and deteriorated performances. One study discovered that the presence of moisture on the interface can affect interfacial adhesion thereby reducing the mechanical performances [43]. Most of the time, delamination happens on poor interface composites. The topography of fiber and matrix decides the condition of the interface. Figure 5 shows the simple schematic view of the fiber/matrix interface.

Stress concentrated at the interface because of two reasons: (1) Different thermal expansion coefficients for fiber and matrix when subjected to thermal processing; (2) different strength properties of both materials. When a low-interfacial adhesive’s composite is subjected to loads, microcracks begin to form at the interface and are propagated to the matrix. Good interface ensures effective load transmitting from the matrix to the fibers, which helps reduce stress concentrations and subsequently improves the overall mechanical properties.

Nano-scale plant fibers, or known as nanocellulose fillers, are getting more and more attention because of its superior reinforcement effects. The high surface area-to-volume ratio of nanocellulose provides a great contact surface between the nanocellulose and the matrix. This created intense interactions at the interface. However, aggregation of nanofillers reversed the strengthening effect. In the well-dispersed nanocomposites, numerous interfacial bonding could be located everywhere inside the matrix even at low filler concentrations. Even distribution of load transmitting among these interfaces in turn yields higher load capability of nanocomposite. 

Interface characterizations could be identified by four (4) methods according to Jose [44] namely, (1) thermodynamic methods (contact angle analysis and gas chromatography), (2) microscopic viewing analysis (scanning electron microscopy (SEM), transmission electron microscopy (TEM), and atomic force microscopy (ATM), (3) spectroscopic techniques (chemical analysis and X-ray diffraction), and (4) micromechanical measurements (single pull-out test). 

### 4.1. Thermodynamic Methods

Wettability analysis is conducted under dynamic conditions where wetting refers to moving of liquid across the surface of the solid and it is a prime factor of superior interface [28]. Topographies of both fibers and matrix influence the wettability. Quantitative measurement on the wettability can be observed from solid–liquid contact angle analysis. This analysis involves dropping a liquid, water in most cases, onto the fiber surface and then the contact angle is calculated, but sometimes other liquids are also applied. 

Better wettability can be attributed to the increased surface roughness and greater exposure of crystalline cellulose by non-cellulosic component removal from the surface, usually by surface chemical treatments [45]. Reduced fiber hydrophilicity and polarity were reported to increase the contact angle with water but found contrary for nonpolar liquids [46]. Polymer matrix are hydrophobic in nature. Fiber with lower hydrophilicity, which is indicated by smaller contact angle, has been observed to have higher compatibility with matrix. Figure 6 shows the condition of wetting with different contact angles, contact angle lower than 30° was claimed to have better interface bonding [47]. On the other hand, viscosity of molten resin, surface conditions, and components may affect the measurement. Jpati and Sengupta [48] investigated the wettability of five (5) plant fibers on different polymer matrixes. Based on the findings, good wettability of fiber/matrix combination were governed by physical and chemical properties of fiber surface and polymer surface tension.

Classical contact angle method provides informative indication of wettability and subsequently the condition of interface adhesion. However, irregular shapes, high batch-to-batch variability of natural fibers, and heterogenous distribution in polymer matrix have made real adhesive properties vary throughout the composite. These analysis limitations can be minimized and overcome by using inverse gas chromatography (IGC). The surface properties of various types of natural fibers could be investigated by employing IGC. IGC is able to determine the surface tension of a variety of natural fibers covering a wide range of cellulose content. 

IGC is a powerful tool that provides important information including dispersive surface energies, γSD, and thermodynamic of adsorption. Gamelas [50] reviewed the capabilities of IGC analysis on cellulose and lignocellulosic materials. Detailed theoretical background is discussed in the review. Plant fibers surface energies and Lewis acid-base interactions are also summarized. Besides, measurement of the surface energies of plant fibers by using IGC with aids of X-ray diffraction and FTIR analysis has proven that plant fibers are potential reinforcements in polymer composites [51]. The increases of γSD are associated with decreases of lignin and/or higher cellulose contents in fiber. Besides, the influences of chemical composition, crystallinity, and morphology of plant fibers were also reported [52]. Furthermore, higher γSD values contributed to more hydrophobic active sites, resulted in better wettability and interface adhesion [53]. On the other hand, Lewis acid-base readings could be used as a guideline to predict the existence of non-cellulosic components on the fiber surface. The removal of non-cellulosic components using alkaline treatment would increase the acidic character of the fibers because of the exposure of predominantly acidic cellulose [54]. Although the interpretation of surface properties in plant fibers by IGC is complex, it is however, a successful technique for characterizing the surface properties of the plant fibers.

### 4.2. Microscopic Viewing Analysis

Microscopic viewing analysis provides concrete graphical evidence in microns level. Topography of fibers enabled us to predict the interface condition and composite’s performances. Various types of viewing tools can be used to analyze the topography conditions as shown in Figure 7.

Scanning electron microscope (SEM) perhaps is the most widely used viewing tool for composite materials. It uses a focused beam of electrons, projected onto the targeted surface. The electrons interact with the surface atoms and produce various signals that contain information about the sample surface topography. A layer of gold was coated on the object surface to ensure stable transmitting of electrons. High definition of surface images allows researchers to identify what had happened on the specimen. Upon comparing treated fiber surface with the raw fiber, subsequent interfacial adhesion and composite characterizations can be predicted. The smooth native fiber surface is mainly due to the waxy layer. Rougher fiber surface and pits were observed after alkaline treatment, which removes non-cellulosic components from the fiber surfaces (Figure 8). This is important for the fiber for mechanical interlocking. Treatment parameters like concentration, treatment time, and temperature normally affect the fiber surface condition [56]. 

On the other hand, SEM analysis can visualize the actual fatigue condition of fiber polymer composite, revealing its interface condition and hence some comments can be deduced [58,59,60,61]. Interface adhesion governing the load transfer mechanism from matrix to fibers, resulted in pulled-out, debonded, or fractured fiber. Voids formation attributed to poor matrix penetration, or insufficient matrix for wetting purpose, can also be found in the SEM micrographs. Embedded fibers in the matrix show good degree of fiber wetting, strengthening the interfacial bonding between the fiber and the matrix. 

Transmission electron microscopy (TEM) analysis is usually conducted on nanoscale fiber reinforcements or film structure composites. Besides, effectiveness of nanocellulose production can be determined using this analysis via nanofiber diameter frequency distributions with aids of computer software [62,63,64]. Consistent nanocellulose fiber dimension typically led to superior composite performances. Besides, nanofiber dispersions and homogeneity are mainly viewed and discussed [65,66]. Fiber agglomeration issue was always found because of poor compatibility between the hydrophilic plant fibers and the hydrophobic matrix. This is more severe in nanocomposites because strong hydrogen bonding between the fibers made them hard to disperse [67]. Hence, a viewing tool like TEM could help in inspection, to locate the agglomeration spots (Figure 9). Fiber agglomeration inhibits complete wetting and to have good interface adhesion [68]. 

Sometimes, atomic forced microscopy (AFM) analysis was used to study the topography of plant fibers. Surface roughness shows explicit information on fiber surface components. High roughness could be associated with the presence of lignin, extractives, and hemicellulose components on the fiber surfaces. Smoother AFM surface observation was due to higher effective fiber surface area for matrix wetting and hence improving fiber/matrix adhesion by mechanical interlocking [70]. Besides, organized scale-like structures could be observed in the AFM image as an indication of cellulose microfibrils existence in the primary cell wall [71]. 

The peak-force quantitative nanomechanical property mapping (PF-QNM) mode is used in order to estimate the indentation modulus of scanned layer. It revealed mechanical characterization and stiffness on selected small scan areas at micro and nanoscale, up to 1 nm^2^ to ensure the indentation analysis was performed on the tip of the nanocellulose fiber surface [72]. Indentation modulus is highly influenced by the anisotropic character of the fibers, which shows the importance of the middle lamella of plant fibers as composite reinforcement [73]. This indentation modulus provides accurate values for shear modulus and torsional strength [74]. 

Regardless of any viewing analysis tools, it is important to visualize the effectiveness of the treatment, actual conditions of the plant fibers, or their interface adhesive conditions. Clean and rough fiber surfaces that are well-dispersed in the polymer matrix always recorded reasonable or superior composite performances. In contrast, poor wettability, improper bonding, fiber breakage, and/or voids reflected bad interfacial adhesion between fiber/matrix. Furthermore, conducting these viewing analyses on nanoscale plant fiber composites are more desired. 

### 4.3. Spectroscopic Techniques

Heterogenous chemical composition and physical dimensional of plant fibers were attributed by numerous factors. Almost all review papers regarding natural plant fibers has mentioning this phenomenon [23,24,75]. The chemical composition of fibers has the most direct relationship with its interface adhesive and characterizations. Each constituent is responsible for different tasks and provides certain tensile reinforcements [76]. Cellulose is the prime contributor to the strength, where hydrogen bonding in the cellulose microfibrils with matrix forms strong interface. Reduction of non-cellulosic constituents could expose celluloses for high bonding intensity, and hence better fiber/matrix interface and load absorbing capabilities. Chemical analysis is another important tool to examine the effectiveness of the treatment by checking the changes of chemical composition in fibers. Chemical modification of fibers is necessary for increased adhesion between hydrophilic fibers and hydrophobic matrix [77]. The most cost-effective and relatively simple treatment is alkaline treatment as it gave positive results in the previous studies. Nevertheless, other treatments are also listed in Table 3. 

X-ray diffraction analysis is used to determine the crystallinity index according to the crystallinity cellulose content in the fibers or fiber polymer composites. Removal of non-cellulosic components will expose and increase the ratio of crystallinity cellulose in the fibers. Cellulose is the best strength tolerator. Hence, higher crystallinity index helps improve mechanical properties of the composite. Cellulose intensity peak designated at around 22°, attributed to (1 1 0) crystallographic plane, is an indication to the crystallinity index value. On the other hand, 16° revealed the cellulose intensity in a particular fiber. When the fiber cellulose contents are high, double peaks may be observed at around 16°, presenting (2 0 0) crystallographic plane [78]. However, for high amounts of amorphous materials including amorphous cellulose, the double peak shows to be smeared, showing a broad peak. Besides, higher crystallite size reduces the chemical reactivity and water absorptivity of the plant fibers [79,80]. Crystallite size increases because of the recrystallization and removal of some crystal defect as the intermolecular force of attraction increases [81].

Fourier transform infrared spectroscopy (FT-IR) is used to identify the chemical functional group that presents in the fiber sample by producing an infrared absorption spectrum. Table 4 shows the peak assignments of lignocellulosic materials. Hydrogen bonding is an important criterion that influences the fiber/matrix interface, thereby affecting the composite properties. It form between hydrogen atom and an electronegative atom, such as fluorine, oxygen, or nitrogen, from another chemical group. Hydroxyl groups in cellulose component forms various kinds of intra- and inter-molecular hydrogen bonding, affecting the physical and mechanical properties [82]. 

Different types of plant fibers consist of various ratios of cellulosic and non-cellulosic constituents. However, with the aid of FTIR, the effects of plant fiber treatments can be quantified with the intensity of absorbance bands. This allowed further interpretation of interface characterizations between plant fibers and matrix. The appearances of new bands in FTIR, may indicate the activation of new bonding especially during the insertion of coupling agents or compatibilizers [83]. The disappearance or diminished absorbance bands reflect the fiber surface components removal upon fiber treatments [84]. The disappearance or diminishing of 1730 cm^−1^ attributed to the carbonylic group C=O stretching vibration of the linkage of carboxylic acid in lignin or ester group in hemicellulose, showing the success of the treatment [85]. Besides, the intensity of C-O stretch peak for acetyl group in lignin at 1237 cm^−1^ is also reduced in treated fibers.

**Table 3 polymers-13-00438-t003:** Chemical composition of raw and treated fiber in previous studies.

Before Treatment	Raw Fiber				After Treatment	Treated Fiber				Ref
	Cellulose (wt%)	Hemicellulose (wt%)	Lignin (wt%)	Wax (wt%)		Cellulose (wt%)	Hemicellulose (wt%)	Lignin (wt%)	Wax (wt%)	
Agave Americana Fibers	68.54	18.41	6.08	0.56	Alkaline	78.65	8.47	4.65	0.46	[86]
					Stearic Acid	81.65	6.31	3.43	0.37	
					Benzoyl peroxide	80.26	7.42	4.33	0.42	
					Potassium permanganate	79.78	6.67	4.10	0.22	
Banyan tree fibers	67.32	13.46	15.62	0.81	5% Alkaline	70.4	10.74	12.7	0.69	[87]
Pennisetum orientale grass	60.3	16	12.45	1.9	HCI acid	56.1	7	5.4	0.3	[88]
					Alkaline	66.7	10.3	8.7	0.7	
Banana fibers	25.51	2.13	42.50	-	KOH	34.24	5.82	2.72	-	[89]
					NaOH	46.46	7.43	5.82	-	
Ramie fibers	73.60	13.81	1.33	0.82	Alkali	90.62	1.04	0.89	0	[90]
					Peroxide	87.43	9.41	0.47	0.10	
					Peroxide and isopropyl alcohol	91.82	5.23	0.28	0.05	
Flax fibers	79.56	8.76	-	-	NaOH/ethanol	87.81	7.48	-	-	[91]
EFB fibers	53.37	19.88	10.74	-	0.8% NaOH	58.93	22.75	7.03	-	[92]
					0.8% Acetic acid	63.21	16.30	7.45	-	

**Table 4 polymers-13-00438-t004:** Peak assignments of the lignocellulosic materials [93].

Band Position, cm^−1^	Assignment
3550–3650	O–H stretching in free or weakly H-bonded hydroxyls
3200–3400	O–H stretching in H-bonded hydroxyls
2840–2940	C–H stretching region
2725	Overtone of interacting C–O stretch and O–H deformation
2568	Overtone of interacting C–O stretch and O–H deformation
1720–1740	C=O stretching in carbonyl
1625–1660	Adsorbed water molecules in non-crystalline cellulose
~1600	Aromatic skeleton ring vibration and vibrations owing to adsorbed water
~1505	Aromatic skeleton ring vibration
1450–1475	C–H deformation and CH_2_ (sym.) + OH deformation
1400–1430	C–H deformation (methoxyl group in lignin)
~1370	C–H deformation (symmetric)
~1327	Syringyl ring breathing with C–O stretching (lignin) and CH2wagging in cellulose
1250–1260	Guaiacyl ring breathing with C–O stretching (lignin)
1240–1245	C–O bond of the acetyl group in xylan and hemicellulose
~1230	Phenolic O–H deformation (lignin)–syringyl structure
1160–1230	C–O stretching of ester groups
1150–1160	C–O–C stretching (anti-symmetrical) in cellulose and aromatic C–H CH_2_ wagging in cellulose
1098–1120	Skeletal vibration involving C–O stretching of the β-glycosidic linkages
~1060	C–OH stretching vibration
1036	Aromatic C–H in plane deformation, guaiacyl and C–O deformation primary alcohol in lignin and C–O stretching in cellulose
1003	Skeletal vibration and C–O stretching in cellulose
890–900	Antisymmetrical stretching owing to b linkage in cellulose
830	Aromatic C–H out of plane vibration owing to lignin

### 4.4. Micromechanical Measurements

Several experimental methods have been used to study the interfacial condition of a natural fiber-reinforced polymer composite, such as micro-indentation test, fiber peel test, single fiber fragmentation test, and single fiber pull-out test. Among these methods, single fiber pull-out test, or just known as pull-out test, is widely applied to analyze the composite interfacial behavior. This review only focused on the pull-out testing for plant fiber polymer composites. 

During the pulling out of a fiber, it experiences several typical stages depending on the interfacial adhesion conditions and the stages are explained in detail in Table 5. Nevertheless, fibers are not always pull-out successfully. Breakage of fibers or matrix being pulled out with fibers might happen. Zhong and Pan [94] studied a simulation with respect to the different fiber pull-out behavior according to the variations of six parameters (interfacial bonding strength, fiber’s strength, matrix’s strength, frictional force between fiber and matrix, embedded length and fiber diameter). All possible pull-out behaviors by different level of interfacial bonding strength are listed in Table 6. In general, higher interfacial bonding strength requires higher tensile loading. However, a large interfacial bonding strength resisting the debonding/slipping of fiber, resulted in fiber breakage, and resulted in transmitting a lower loading.

Although, a few theoretical modellings have been developed, yet none of them could possibly 100% represent the actual natural fiber pull-out. This is because plant fibers are not always in perfect cylindrical shape as synthetic fibers. Heterogenous dimension of plant fibers disturbed effective load-transferring mechanism. An irregular shaped fiber may debond at lower loads, and split from each other, resulting in fibrillation. In general, fibrillation (splitting fibers into finer fibrils) increases the total fiber surface area for better adhesion in composites. However, lower bonding strength was found for each fibril, resulting in lower fiber pull-out capability. Non-consistent fiber diameter throughout the fiber was marked, yet no discussion on this matter was found in the previous studies [96,97]. 

## 5. Importance of Interfacial Adhesion Condition on Plant Fiber Polymer Composites Performances

Interfacial adhesion of the fiber/matrix plays a vital role in determining the physical, mechanical, and thermal properties of its composites. Ideally, two materials with similar properties should be combined. For example, in order to create superior interfacial adhesion and strong bond, hydrophilic and hydrophilic materials should be used, and vice versa. Better dimensional stability could be attained when hydrophobic fillers and hydrophobic matrices are combined. Unfortunately, in the case of plant fiber polymer composites, the combination of hydrophilic plant fibers and hydrophobic polymer matrix can lead to inferior dimensional stability as a result of poor interfacial adhesion. However, enhancement could be done through treatment. 

Impurities removal through effective treatment has made fibers rearrange themselves in a more compact manner, creating stronger composites [98,99]. This manner has changed the physical characteristics of plant fibers and its composites. Besides, the bonding strength at the interface regulates the proper load-transferring mechanism which in turn to bestow the composite with high toleration toward maximum load bearing. On the other hand, breaking a high interfacial bonding strength requires relatively higher amount of thermal energy, making it perform well under elevated temperatures. 

Fiber treatments are often conducted to improve the interfacial adhesion. Optimum treatment parameters resulted in a higher resistance to the pull-out process. Better interfacial adhesion and penetration of molten polymer into rough fiber surface lead to a better mechanical interlocking [100]. Contrarily, poor interface showing fiber debonding and fiber pull-out for under- and over-treated fibers, resulted in poor composite characteristics. Hence, interfacial adhesion condition on plant-fiber-reinforced polymer composite is a crucial factor to control its performances.

### 5.1. Physical Properties

The insertion of natural fibers in PLA composites have induced a non-negligible pro-degradative effect on PLA molten state. The higher pace of hydrolysis and biodegradation of PLA polymer was found with increased fiber contents, thereby reducing complex viscosity of molten state [101]. Fortunately, better interfacial bonding adhesion between fiber/matrix minimizes this drawback and retains the viscosity close to pure PLA. This unchanged viscosity value allows plant-fiber-reinforced polymer composites to fabricate with similar processing parameters as pure polymer. Manufacturers may introduce plant fiber into polymer products without varying the processing setting.

The fiber/matrix adhesion condition is an important criterion in determining the water absorption behavior. The main constituents of natural fiber are cellulose and hemicellulose, which was dominated by hydroxyl and carboxyl groups. Owing to its easy attachment to water molecules via hydrogen bonding, these functional groups are hydrophilic in nature. Mildly alkaline sodium bicarbonate treatment has reported good reduction of hemicellulose and lignin contents in natural fibers [102]. However, it activates and worsens the propagation of damage phenomena, resulting in higher water absorption [103].

Alkaline treatment has helped in improving the mechanical interlocking and chemical bonding between fibers and matrix, resulting in superior properties. Kenaf fibers have relatively higher non-cellulosic components than hemp fibers. This made the removal of impurities for kenaf fibers effective, indicating higher bulk density changes and performance improvement [104]. Sodium bicarbonate treatment found insignificant changes in fiber density [100]. However, smaller diameter of treated fiber increases corresponding aspect ratio (L/d) values. This increment is one of the most influential parameters in affecting the mechanical properties. However, Madhu et al. 2020, found no correlation relationship between fiber diameter and mechanical properties in their study [86]. 

### 5.2. Mechanical Properties

Plasma treatment has been accounted for the creation of rougher and higher polarity fiber surface and hence led to better interface adhesion and tensile strength [105]. Several fiber treatments also were conducted and compared. All treated fibers showed higher tensile and flexural strength but lower impact performance [106]. However, deterioration of tensile strength was reported when excessive treatment parameters were adopted, as they damage the crystalline cellulose structure [107,108]. Chen et al. [109] formed a polymerized epoxidized soybean oil interfacial layer at the interface between bamboo fibers and poly(lactic acid) (PLA) matrix using in situ polymerization. The cationic polymerization of soybean oil was found to be initiated at the interface, linking both the fiber and the matrix. The tensile strength, tensile modulus, fracture elongation, and storage modulus of the biocomposites were significantly enhanced. 

It is worth highlighting that the interfacial adhesion quality affects the flexural strength more than the tensile strength [110]. Flexural bending loads imply a combined compressive/tensile and interfacial shear stress to the specimens, in which interfacial adhesion plays a fundamental role. Kafi et al. [70] recorded a 45.10% improvement in flexural strength for plasma-treated jute composites [70]. Positive correlation of work of adhesion (Wa) with flexural strength in longitudinal and transversal direction was observed [111]. Besides, poor interfacial adhesion between fibers and matrix resulted in inferior impact strength behavior. Complete fiber pull-out and debonding absorbed large amounts of energy, and this could be visualized by using SEM viewing tool. Fiber fracture in proper interface requires less energy, hence treated fiber composite usually resulted in lower impact values.

### 5.3. Thermal Properties

The fiber–matrix interface may play a vital role in the process of the decomposition and degradation of plant fiber composites. Thermal degradation started with thermal energy absorption to break the bonding. Greater thermal stability is attributed to the larger activation energy, manifested by outstanding interface [112,113]. In thermogravimetric analysis (TGA), a lower initial and second mass loss during pyrolysis was attributed to the fewer moisture contents bounded and low hemicellulose and lignin components, respectively, under strong interfaces [114]. The third and most prominent mass loss stage was attributed to the thermal degradation of cellulose component. Hence, higher thermal decomposition temperature was recorded for treated plant fiber composites, having good interfacial bonding. However, agglomeration of fiber reduces the fiber/matrix adhesion intensity, thereby reducing the thermal stability of the composite [115]. 

Dynamic mechanical analysis (DMA) of composites records the energy dissipation during relaxing and compression states (which similar to flexural analysis), throughout a range of temperature. Weaker interface adhesion condition expects higher energy dissipation by internal friction [116]. Hence, higher storage modulus retained for treated plant-fiber-reinforced polymer composites, with lower loss modulus. Molecular motions at the interfacial region are responsible for the composite’s damping behavior. Amorphous PLA matrix shows relatively higher damping magnitude, but this can be reduced by plant fiber reinforcements. An ideal interfacial adhesion facilitates load-transferring mechanism but not contributing to damping mechanism, according to composite damping rule [117],
$$\tan\delta_{C}^{0}=\left(1-V_{f}\right)\tan\delta_{m}$$
where *V*_f_ is the volume fraction of fibers, tan δ_m_ is the damping of the matrix. 

Reduced polymer chains’ mobility due to premium interfacial bonding reduces damping values in all temperature ranges. Insertion of plant fibers has reported greatly improved the damping and storage modulus values, as higher linkages are found for fibers-reinforced polymer composites [118]. 

On the other hand, insertion of plant fibers offers a nucleating site for the cold crystallization of polymer [119]. This implies rapid polymer crystallization, resulting in higher crystalline temperature (earlier crystalline during cooling). However, the immobilization effect, which refers to polymer chain mobility restriction due to appearance of plant fiber, become more dominant upon higher fiber contents [120]. Besides, glass transition temperature, *T*_g_, refers to the temperature in which the polymer chains start to move. At this temperature, amorphous regions of semicrystalline polymer become slippery, experiencing a transition from glassy state to rubbery state. A strong interfacial bonding delayed the transition process, by shifting to higher *T*_g_ values [121,122]. However, thermosetting polymer is cured and hardened by extensive cross-linking, creating strong polymer. Therefore, insertion of plant fibers and interface condition of thermoset composite do not alter the *T*_g_ values [123,124]. 

## 6. Conclusions

Natural plant fibers have been recognized as a promising candidate in reinforcing and enhancing the properties of polymeric composites. Plant fibers possess several advantages such as readily availability, low cost, high strength-to-weight ratio, volume-to-weight ratio, and excellent biodegradability. These characteristics have perpetually raised the reputation of plant fibers as an ideal filler for polymeric composites. However, its hydrophilic nature has inhibited the compatibility with hydrophobic polymer. This paper reviewed the fiber/matrix interfacial bonding mechanisms as good interlayer will form strong linkages and enable maximum stress transmission between fibers and matrix. Four types of interfacial bonding mechanisms, namely physical adhesion, electrostatic adhesion, chemical adhesion, and mechanical interlocking and their respective effects on the properties of composites were discussed. It was found that the interfacial adhesion is a result of the mixture of these bonding mechanisms. However, one of them usually plays a dominant role. Quantification and analysis of interfacial adhesion properties are very important to compare or predict the properties of plant-fibers-reinforced polymer composites. The common interface characterization methods including thermodynamic methods, microscopic viewing analysis, spectroscopic techniques, and micromechanical measurements have been highlighted and compared. Although this topic has not been extensively studied, it however provides a meaningful insight into characterizing the performance of the plant-fibers-reinforced polymer composites. In the near future, more studies on the relevant topic are anticipated to overcome the bottleneck in creating greater fibers/matrix compatibility.

## Figures and Tables

**Figure 1 polymers-13-00438-f001:**
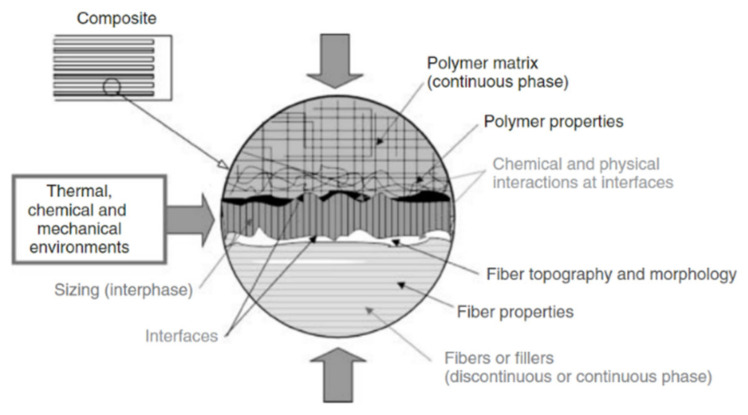
Schematic illustrating interface of fiber/matrix composite. Reproduced with permission from [12]. Copyright 2013 John Wiley and Sons.

**Figure 3 polymers-13-00438-f003:**
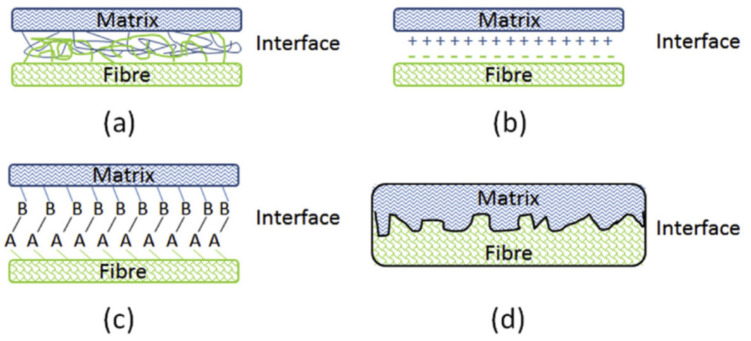
Schematic figure of fibers/matrix interfacial bonding mechanisms with the methods of (**a**) interdiffusion, (**b**) electrostatic adhesion, (**c**) chemical bonding, and (**d**) mechanical interlocking. Reproduced with permission from [27]. Copyright 2016 Elsevier.

**Figure 4 polymers-13-00438-f004:**
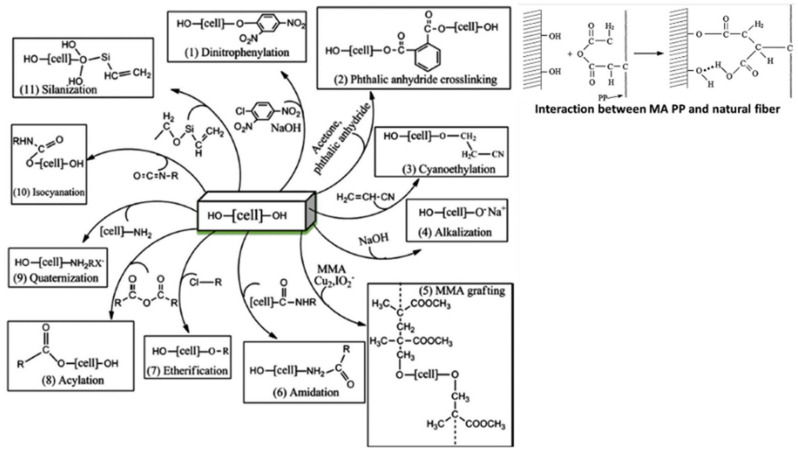
Chemical modification treatments on fiber’s surface. Reproduced with permission from [40]. Copyright 2019 Elsevier.

**Figure 5 polymers-13-00438-f005:**
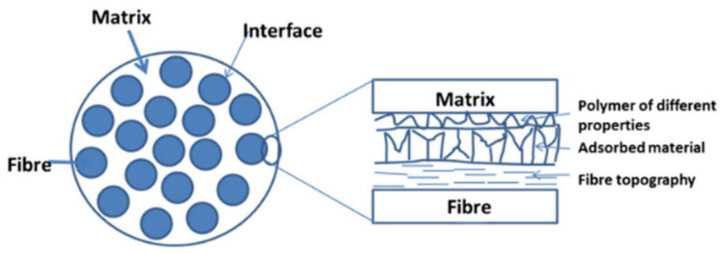
Simple schematic of plant fiber/matrix interface. Reproduced with permission from [43]. Copyright 2015 Elsevier.

**Figure 6 polymers-13-00438-f006:**
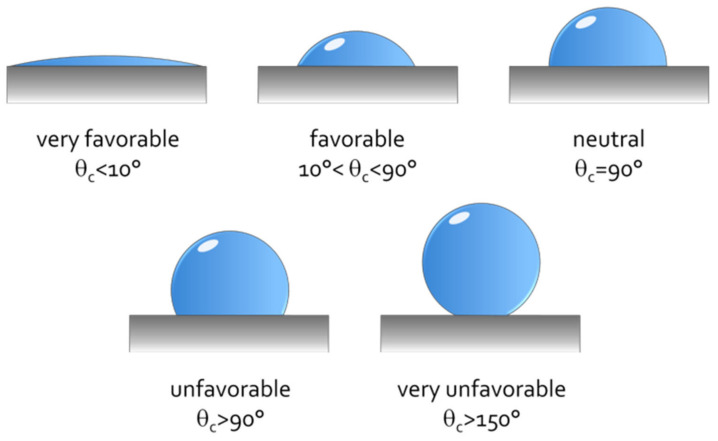
Wettability condition of different contact angles. Reproduced with permission from [49]. Copyright 2020 University of Stuttgart.

**Figure 7 polymers-13-00438-f007:**
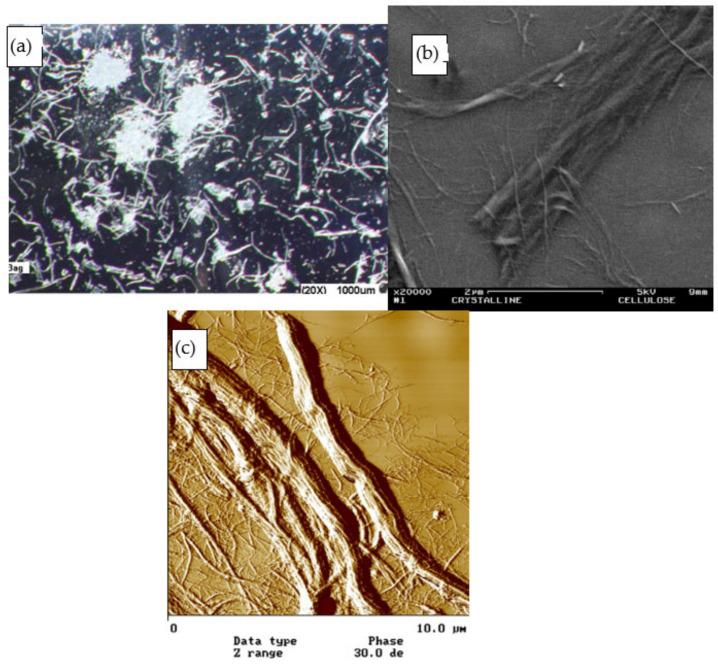
(**a**) Optical, (**b**) scanning electron microscope (SEM), and (**c**) atomic forced microscopy (AFM) micrographics of cellulose microfibers obtained from bagasse. Reproduced with permission [55]. Copyright 2008 Elsevier.

**Figure 8 polymers-13-00438-f008:**
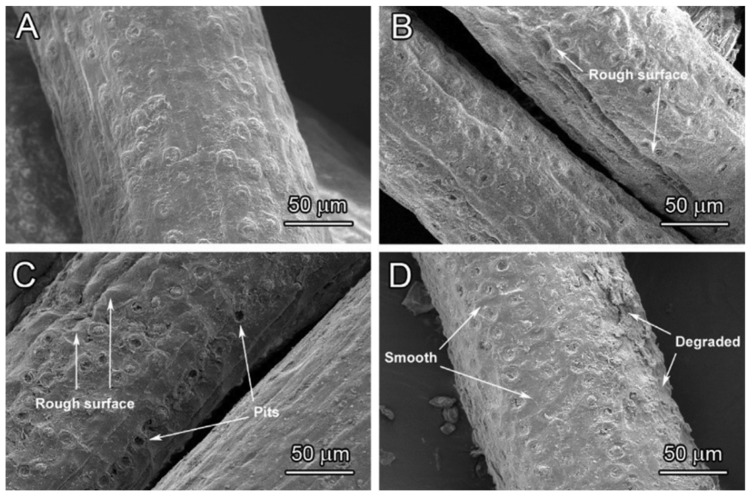
SEM surface analysis of (**a**) raw coir fiber (**b**) 24 h, (**c**) 96 h, and (**d**) 168 h treated coir fibers. Reproduced with permission from [57]. Copyright 2018 Elsevier.

**Figure 9 polymers-13-00438-f009:**
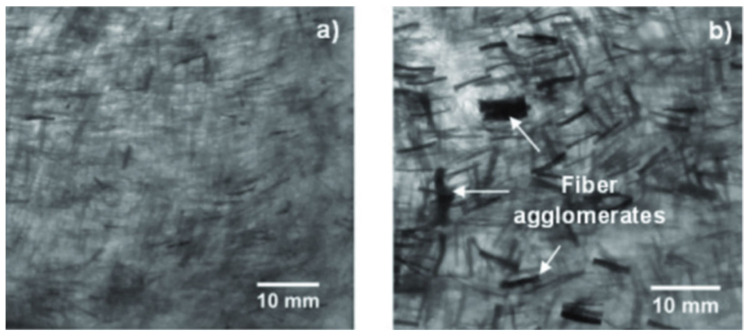
TEM images of 8 wt% of nanocellulose from pineapple leaf; (**a**) raw and (**b**) treated fibers, reinforced in PP composite. Reproduced with permission from [69]. Copyright 2016 SciELO.

**Table 2 polymers-13-00438-t002:** Chemical composition of frequently used plant fiber [24].

Fibers	Cellulose (wt%)	Hemicellulose (wt%)	Lignin (wt%)	Waxes (wt%)
Bagasse	55.2	16.8	25.3	-
Bamboo	26–43	30	21–31	-
Flax	71	18.6–20.6	2.2	1.5
Kenaf	72	20.3	9	-
Jute	61–71	14–20	12–13	0.5
Hemp	68	15	10	0.8
Ramie	68.6–76.2	13–16	0.6–0.7	0.3
Abaca	56–63	20–25	7–9	3
Sisal	65	12	9.9	2
Coir	32–43	0.15–0.25	40–45	-
Oil palm	65	-	29	-
Pineapple	81	-	12.7	-
Curaua	73.6	9.9	7.5	-
Wheat straw	38–45	15–31	12–20	-
Rice husk	35–45	19–25	20	-
Rice straw	41–57	33	8–19	8–38

**Table 5 polymers-13-00438-t005:** Short fiber pull-out stages [95].

Steps	Fiber Pull-Out Stages
1	Initiation of interfacial microfailure at fiber tips due to tensile stress concentration in matrix around fiber tips: from about 50% of ultimate load
2	Separation at the interface, formation of a microvoid.
3	Propagation of interfacial microfailures along fiber sides due to critical shear stress concentration: from about 75% of ultimate load; a fringe pattern of shear mode and microcracks are observed in the matrix along fiber sides.
4	Occurrence of plastic deformation bands in the matrix due to stress concentration caused by the reduction of fiber load bearing capability; crack opening and slow crack propagation through plastic deformation bands (ductile crack propagation).
5	Brittle crack propagation: when crack size reaches a critical value (about 1 mm), they propagate along fiber sides and through the matrix, which leads to composite failure.

**Table 6 polymers-13-00438-t006:** Possible pull-out conditions due to different level of interfacial bonding strength [94].

	Possible Pull-Out Conditions
(a) Low Interfacial Bonding Strength	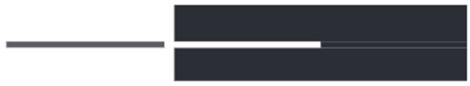
(b) Medium Interfacial Bonding Strength	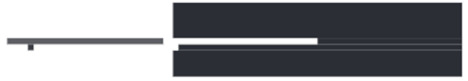
(c) High Interfacial Bonding Strength	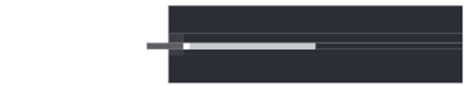

## Data Availability

Not applicable.

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
