# Peer review of "Importance of Interfacial Adhesion Condition on Characterization of Plant-Fiber-Reinforced Polymer Composites: A Review"

_polymers, 2021, doi:10.3390/polym13030438_

Round 1

Reviewer 1 Report

The paper “Importance of Interfacial Adhesion Condition on Characterization of Plant Fiber Reinforced Polymer Composites: A review”  gives a wide review of the importance of interfacial adhesion conditions on the properties of plant fiber reinforced polymer composites. Some minor changes are needed for publication:

Introduction

Line 47: Please, change “is” by “are”

Natural Plant Fibers

Line 126: Please delete 0 from title or introduce it in Introduction Section, following journal template.

Quantifying and Analysis of Interfacial Adhesion Condition

Figure 6: Please consider to reduce the size of the picture. If the angle is 10°, Is the wettability condition very favorable or favorable? What happens between 90° and 150°?

Figure 7. Please clarify in the picture which corresponds with a), b) and c).

Table 5. It is possible to put “Steps” in one line?

Line 485: Can give the authors give an idea about how the theoretical modellings can be improved to achieve better results in the natural fiber pull-out?

Line 583. According the journal template, it is necessary to introduce a reference for the equations?

Line 584. Please change”Where” by “where”

Author contributions. Please add the real author contributions to this section.

Bibliography. Please add DOI when it is possible.  

Author Response

  1. Line 47: Please, change “is” by “are”

The word has been revised accordingly.

  1. Line 126: Please delete 0 from title or introduce it in Introduction Section, following journal template.

The 0 removal on the title numbering have been removed.

  1. Figure 6: Please consider to reduce the size of the picture. If the angle is 10°, Is the wettability condition very favorable or favorable? What happens between 90° and 150°?

The size of the figure has been reduced.

The term of very favorable and favorable are Grammarly correct, please check the oxford online dictionaries link (https://www.oxfordlearnersdictionaries.com/definition/american_english/favorable), the term “very” in the figure was meant to show the difference between two wetting angles. Similar scenario for 90o and 150o.

  1. Figure 7. Please clarify in the picture which corresponds with a), b) and c).

The numbering on the figure have been inserted.

  1. Table 5. It is possible to put “Steps” in one line?

We do not quite sure what is the meaning of putting the steps in one line.

  1. Line 485: Can give the authors give an idea about how the theoretical modellings can be improved to achieve better results in the natural fiber pull-out?

Theoretical modelling is similar to other simulation modelling, allow us to predict the outcomes. It provides us the best dimension of natural fibre in order to achieve optimum pull out strength. From there, fiber treatments and/or modification may introduce to obtain similar dimension, reducing intense fundamental studies.

  1. Line 583. According the journal template, it is necessary to introduce a reference for the equations?

The reference has been inserted into manuscript.

  1. Line 584. Please change”Where” by “where”

The word has been revised accordingly.

  1. Author contributions. Please add the real author contributions to this section.

The section of author contributions has been filled up.

  1. Please add DOI when it is possible.  

The bibliography references were created by using Endnote software. All the references are downloaded from official journal website. Hence, we found difficulties when the official website didn’t provide the DOI details in its reference.

Reviewer 2 Report

In general the Authors did a good job with the review, but there are some issues that should be adressed in my opinion.

1. English language should be corrected.

2. Considering the application of natural plant fibers. Authors are referring to the "green materials". Are the natural fibers indisputably green? When the waste materials are used, definitely. But what about the cultivation aimed at fiber manufacturing, water use, land degradation? This issue is discussed in the literature over the last years and should be adressed here. Please check the papers: - https://doi.org/10.1016/j.compositesa.2003.09.016 - https://doi.org/10.1016/j.jclepro.2014.03.017 - https://doi.org/10.3390/ma13163541 - https://doi.org/10.1007/978-3-319-13847-3_6 - https://doi.org/10.3390/polym11111791

3. The sections related to the interface mechanisms should be enhanced. Considering chemical adhesion and Figure 4, it should be improved and the chemical reactions should be given. Also I miss the very valuable papers from the last years related to the natural fiber composites, which discuss the issue of interface: - https://doi.org/10.1016/j.matpr.2017.11.276 - https://doi.org/10.1016/j.ejpe.2017.11.005 - https://doi.org/10.1177/0731684418756762 - https://doi.org/10.1007/s00226-020-01203-3 - https://doi.org/10.1016/j.jclepro.2017.10.101

4. Generally the discussion should be more comprehensive, since the topic is very popular among the researchers all over the world.

Author Response

  1. English language should be corrected.

The English language in the manuscript has been checked carefully and improved.

  1. Considering the application of natural plant fibers. Authors are referring to the "green materials". Are the natural fibers indisputably green? When the waste materials are used, definitely. But what about the cultivation aimed at fiber manufacturing, water use, land degradation? This issue is discussed in the literature over the last years and should be addressed here. Please check the papers:

(1) https://doi.org/10.1016/j.compositesa.2003.09.016

(2) https://doi.org/10.1016/j.jclepro.2014.03.017

(3) https://doi.org/10.3390/ma13163541

(4) https://doi.org/10.1007/978-3-319-13847-3_6

(5) https://doi.org/10.3390/polym11111791

Thank you very much on raising such important point on natural plant fibers polymer composites production and I had read all the given manuscripts on above.

I am totally agreed natural fiber polymer composites will produce wastes in some forms. However, I would like to defend on natural fiber composite as its life cycle assessment did have better results compared to synthetic fiber composites, although in the unsatisfactory manner.

However, large amount of wastewater created during natural fiber production could be reduced by using enzymatic retting process, as shown in one of my previous manuscript (https://doi.org/10.1155/2020/6074063). Secondly, commercialize plantation of kenaf plant could reduce the dependence of wood made papers, indirectly reducing illegal logging activities. Besides, empty fruit bunch, coir, banana pineapple leaf natural fiber are side products of the commercial plantations, which may disposed with open burning. And lastly, life cycle assessment in above manuscript did not take account carbon locking via photosynthesis during plantation. Furthermore, production of synthetic fibers also creates certain wastes.

Therefore, in my opinions, natural fiber is still an excellent option on polymer composites.

  1. The sections related to the interface mechanisms should be enhanced. Considering chemical adhesion and Figure 4, it should be improved, and the chemical reactions should be given. Also I miss the very valuable papers from the last years related to the natural fiber composites, which discuss the issue of interface:

https://doi.org/10.1016/j.matpr.2017.11.276

https://doi.org/10.1016/j.ejpe.2017.11.005

https://doi.org/10.1177/0731684418756762

https://doi.org/10.1007/s00226-020-01203-3

https://doi.org/10.1016/j.jclepro.2017.10.101

Thank you for your suggestions, the section has been revised. Since too many reactions were involved. Therefore, we mentioned and cited the reference of natural fiber chemical treatment. Anyone who likes to understand further the chemical reaction may follow the citations.

  1. Generally the discussion should be more comprehensive, since the topic is very popular among the researchers all over the world.

The discussion has been done in a more comprehensively manner to better reflect the core idea of this review.